# The Complex Cyto-Molecular Landscape of Thyroid Nodules in Pediatrics

**DOI:** 10.3390/cancers15072039

**Published:** 2023-03-29

**Authors:** Davide Seminati, Stefano Ceola, Angela Ida Pincelli, Davide Leni, Andrea Gatti, Mattia Garancini, Vincenzo L’Imperio, Alessandro Cattoni, Fabio Pagni

**Affiliations:** 1Department of Medicine and Surgery, Pathology, Fondazione IRCCS San Gerardo dei Tintori, 20900 Monza, Italy; 2Department of Endocrinology, Fondazione IRCCS San Gerardo dei Tintori, 20900 Monza, Italy; 3Department of Radiology, Fondazione IRCCS San Gerardo dei Tintori, 20900 Monza, Italy; 4Department of Surgery, Fondazione IRCCS San Gerardo dei Tintori, 20900 Monza, Italy; 5Department of Pediatrics, Fondazione IRCCS San Gerardo dei Tintori, 20900 Monza, Italy

**Keywords:** thyroid nodule, pediatrics, fine-needle aspiration, ultrasound, molecular pathology

## Abstract

**Simple Summary:**

Pediatric thyroid ultrasound-guided fine-needle aspiration includes the use of dedicated ultrasound guidance algorithms to improve the accuracy and increase the diagnostic yield of the procedure. The refinement of imaging criteria and risk stratification can potentially spare cytological assessment of benign nodules. Molecular testing can also be used to differentiate benign from malignant lesions and to guide their further management. The assistance of artificial intelligence tools will be helpful in this investigation.

**Abstract:**

Thyroid fine-needle aspiration (FNA) is a commonly used diagnostic cytological procedure in pediatric patients for the evaluation of thyroid nodules, triaging them for the detection of thyroid cancer. In recent years, greater attention has been paid to thyroid FNA in this setting, including the use of updated ultrasound score algorithms to improve accuracy and yield, especially considering the theoretically higher risk of malignancy of these lesions compared with the adult population, as well as to minimize patient discomfort. Moreover, molecular genetic testing for thyroid disease is an expanding field of research that could aid in distinguishing benign from cancerous nodules and assist in determining their clinical management. Finally, artificial intelligence tools can help in this task by performing a comprehensive analysis of all the obtained data. These advancements have led to greater reliance on FNA as a first-line diagnostic tool for pediatric thyroid disease. This review article provides an overview of these recent developments and their impact on the diagnosis and management of thyroid nodules in children.

## 1. Introduction

In recent years, the overall incidence of thyroid malignancy has increased mainly due to significant improvements in high-resolution ultrasonography (US)-guided fine-needle aspiration (FNA), leading to earlier cancer detection [1]. US is a non-invasive imaging method that is suitable for thyroid evaluation and is well accepted by patients of all ages, whereas FNA is considered the best method for evaluating thyroid nodules to minimize unnecessary surgeries or establish the proper extent of surgery [2,3]. Histological assessment remains the reference standard for thyroid nodule diagnosis [4]. According to the Surveillance, Epidemiology, and End Results (SEER) database, thyroid cancer cases in individuals under 20 years old account for only 2.3% of all thyroid cancer diagnoses, but they still represent the second most common adolescent malignant neoplasms, sharing a relatively low mortality with their adult counterparts even in advanced forms (globally <2%), although the rarity of these conditions in children makes data collection challenging [5,6,7,8,9,10]. Currently, there is no consensus on the age upper cut-off for defining the pediatric group for thyroid diseases, with different options proposed by the World Health Organization (19 years old), the American Thyroid Association (18 years), and the American Academy of Pediatrics (21 years), with some studies suggesting 14 or 22 years old cut-off for a better prognostic value [10].

Substantial differences in the clinical, pathological and molecular aspects of pediatric thyroid cancers make their preoperative assessment more challenging. From a clinical standpoint, pediatric nodules are relatively less frequent (0.2–5% vs. 20–70%), but they have a greater risk of malignancy (mean ROM, 19–26% vs. 5–15%) and cancer prevalence (estimated from 1.6-fold to 2.5-fold higher), with the peak of incidence being among 15–19 years old [2,3,4,6,7,10,11,12,13,14]. Moreover, pediatric patients often present with occult diffuse infiltrative lesions and extrathyroidal extension (e.g., metastases in regional lymph nodes or pulmonary metastases), stressing the role of non-invasive methods (US + FNA) for their correct assessment [3,6]. They are more often associated with female sex (especially in the post-pubertal >14 years age group, probably due to different endocrine, metabolic and immune characteristics), Asian origins, radiation exposure and inherited syndromes (5–15% of cases) [7,9,10,15]. Correctly recognizing malignant nodules in children is essential because of their greater rate of complications related to surgical treatment, with potential long-term impact on growth and bone health, and for their better response to radioactive iodine therapy (RAI) [2,10]. For preoperative assessment, the commonly used US size cut-offs for risk stratification of thyroid nodules may not be adequate for the age thyroid volume; moreover, the molecular landscape of the most frequent histotype, papillary thyroid carcinoma (PTC), representing approximately 80–90% of cases, is characterized by lower rates of BRAF p.V600E mutation and higher rates of RET/PTC translocations, as compared to what is described in the adult counterpart, and 15–40% of PTC cases are of histological high-risk variants (e.g., tall cell, diffuse sclerosing and solid/trabecular variants) [3,7,9,10,14,15]. As in adults, the second and third most frequent histotypes are follicular thyroid carcinoma (FTC) and medullary thyroid carcinoma (MTC), respectively, while poorly differentiated thyroid carcinoma (PDTC) and anaplastic thyroid carcinoma (ATC) are particularly rare entities [10].

For these reasons, a more comprehensive understanding of the thyroid cancer background in pediatrics is mandatory to ensure their best management. This review will provide an overview of the current state of the field and the various new methods and technologies that are being used to improve the accuracy of thyroid FNA in this population.

## 2. Materials and Methods

We focus on the most recent literature to provide an update on the advancements in the thyroid nodules’ assessment in pediatrics, performing a literature search (date of search: 1 January 2023) in PubMed (Medline) for studies published from 1 January 2020 to 31 December 2022 using the following search terms: “thyroid FNA” and “pediatric(s)” or “children”, requiring the terms “pediatric(s)” or “children” to appear in the title. To further expand the information to articles published before the censoring period, a secondary search of the literature was manually conducted from the references of our primary search, including papers by the application of the same inclusion and exclusion criteria. Articles satisfying the following inclusion criteria were included in our review (regardless of the study design): [1] study was written in English; [2] the full article could be obtained. Articles satisfying the following exclusion criteria were excluded from our review: [1] study was written in a non-English language; [2] the full article was not available; [3] study was not related to thyroid FNA in pediatrics; [4] study was not published in a peer-reviewed journal. Literature review and data extraction were conducted independently by two reviewers (D.S. and F.P.). Doubts or disagreements regarding the inclusion or exclusion of manuscripts were resolved through discussion between the reviewers until a consensus was reached (search strategy summary in Table 1).

## 3. Discussion

### 3.1. Ultrasound Assessment of Thyroid Nodules in Pediatrics

Worldwide, the most employed ultrasound Thyroid Imaging Reporting and Data Systems (TI-RADS) are the American College of Radiology (ACR) TI-RADS, European Thyroid Association (EU) TI-RADS and American Thyroid Association (ATA) guidelines, with known limits in identifying malignant tumors other than PTC [3,15,16,17]. ACR and EU TI-RADS have been successfully tested and implemented in adult practice, the former because of the simplicity and effectiveness of the scoring system, the latter for its pattern recognition approach, but both have not been well studied in children [18]. Furthermore, some authors blame all classification systems for their high missed malignancy rate (approximately 50%), supposedly due to the employment of non-optimal size cut-offs for FNA selection, resulting in a high rate of benign FNA (up to 80%), or either for the non-recognition of unique imaging pediatric characteristics, such as ectopic intrathyroidal thymic tissue (2–5%) resembling microcalcifications [2,3,6,7,18,19,20,21]. Currently, ATA guidance is the only published guideline for the management of thyroid nodules harboring a section explicitly dedicated to the pediatric population [15]. According to ATA guidelines, US-FNA in children is indicated for all solid or partly solid nodules greater than 1 cm in size regardless of their other echographic features (Figure 1A–E) [15]. US-FNA is also recommended for all nodules smaller than 1 cm harboring suspicious features or clinical risk factors, such as Hashimoto thyroiditis, which involves approximately 20–25% of malignant pediatric cases vs. <5% of benign nodules (Table 2) [2,6,15].

This differs from what is applied to the adult population by all the systems, with FNA performed only once the nodules are ≥1 cm in size and harbor suspicious US features (Table 3).

In this direction, Dunya et al. investigated 77 pediatric FNA, half of which had suspicious or malignant cytology findings. Using ATA guidelines rather than the ACR TI-RADS system, they observed a better sensitivity but a worse positive predictive value [7]. Indeed, the ATA system tends to produce a higher number of false-positive results than the ACR system, even if both demonstrated only modest overall diagnostic performance [12]. A slightly superior impact was noted for ACR TI-RADS in the setting of indeterminate class III Bethesda System for Reporting Thyroid Cytopathology (TBSRTC) nodules, due to its ability to appropriately select those nodules that require surgery [22]. According to the ATA, hyperfunctioning lesions are usually resected surgically and do not necessarily require FNA [7,15]. Finally, in the setting of autoimmune thyroiditis, US should be performed, especially in children with other clinical risk factors, considering that PTCs arisen in this context harbor a favorable outcome despite their more frequent multifocal pattern of growth (Table 2) [2]. At the same time, there is no evidence suggesting increased aggressiveness for malignancies arising in genetic cancer syndromes, so prophylactic thyroidectomy is no longer recommended, and US surveillance every 1–3 years is indicated [2]. Nonetheless, prophylactic surgery remains recommended in children harboring RET mutations (MEN syndromes) and at risk of medullary thyroid carcinoma (MTC) onset, with the timing of surgery determined by the known progression risk of every specific mutated codon [2]. However, the US-based surveillance of nodule growth did not prove reliable in assessing the malignant nature of children’s lesions, given the particularly slow growth of many thyroid cancers [1,14].

### 3.2. Fine-Needle Aspiration (FNA) of Thyroid Nodules in Pediatrics

The classification of thyroid FNA samples using The Bethesda System for Reporting Thyroid Cytopathology (TBSRTC) is recommended for children by ATA guidance, which is characterized by more aggressive criteria for FNA than in adults [6,23]. Furthermore, all the systems suffer from a lack of well-defined FNA risk of malignancy (ROM) for pediatric nodules, especially in the indeterminate categories (TBSRTC classes III–IV), although the ultrasound features predictive of malignancy coincide with those of adults [7,14,20]. However, the implementation of Rapid On-Site Evaluation (ROSE) has proven effective in reducing the non-diagnostic rate in the pediatric cohort, on average, halving it [20,24]. Based on the generally literature-recognized higher overall ROM in children, ATA guidelines suggest that all FNA positive and indeterminate nodules should undergo surgical excision and comprehensive cervical lymph nodes US examination (Table 4 and Table 5) [4,15].

The assessment of ROM in a single cytological category and its comparison with its adult counterpart has been extensively explored in recent years, with mixed results. Higher ROM for pediatric patients has been described in the setting of Bethesda classes III, IV and V by Dunya et al., and a study on 340 pediatric FNA confirmed this finding, with greater TBSRTC expected ROM in classes ≥III, suggesting that FNA should be performed for all nodules with ACR TI-RADS score ≥4, independently of their size [7,31]. Single reports even suggest differential ROM depending on the side of the gland affected by nodular disease (e.g., right vs. left lobe) [21]. In contrast, an extensive meta-analysis involving 3687 pediatric and 145,066 adult patients with thyroid nodules showed a comparable frequency and ROM for benign (class II) and indeterminate (classes III and IV) categories, with a relatively higher rate of diagnostic resection for indeterminate classes in the pediatric population, suggesting the risk of overtreatment based on comparable ROM with the adult counterpart [41]. However, possible selection biases can explain this underestimation of ROM in pediatric patients, accounting for a potential increase in the relative risk from 1.6-fold to 2.5-fold compared to adults [12]. A more detailed analysis investigated the relationship between cytological atypia (e.g., nuclear or architectural only, nuclear and architectural at the same time, oncocytic atypia) and the final ROM of class III, revealing a more prominent impact of nuclear features on the final ROM, irrespective of the coexistence of architectural atypia [42]. Moreover, this study further stressed the possible role of repeat biopsy in indeterminate cases of nuclear-only atypia, suggesting a more conservative approach than the ATA guidelines [42]. Finally, regardless of the generally higher ROM, the overall comparable rate of indeterminate FNA diagnoses in children and adults should rule out the possibility that pathologists may fear overdiagnosis in young patients, without falling back on the comfortable security of Bethesda classes III-IV categories, prompting just US follow-up or FNA repetition.

### 3.3. Molecular Testing in Pediatrics

Thyroid molecular genetic testing offers the possibility of obtaining additional information to assist in treatment decisions, particularly for indeterminate nodules at FNA, helping to guide the choice of surgery or observation, even if ATA still does not recommend such analyses on cytological pediatric specimens [2]. To date, only Next-Generation Sequencing (NGS) tests have been validated in children, although large comprehensive studies are lacking, while gene expression classifier tools have not yet been extensively tested and validated [2,8]. Some studies have suggested that microRNA (miRNA) pathways in children are similar to those in adults and that the combination of NGS analysis with miRNA expression raised the diagnostic accuracy in some cases [43]. Children have a high detection rate (approximately 80–90%) of pathogenic variants in differentiated thyroid cancers (DTC) and are more likely to have gene rearrangements rather than point mutations, among which DICER1 variants are more common than in adults (Table 6) [8,44,45]. In particular, about 30% of pediatric cancers harbor BRAF/RAS mutations, while gene fusions are present in up to 60% of cases, mainly in younger patients with PTC, usually involving RET or NTRK1/3, as also ALK, BRAF and PPARG [8]. Overall, similar survival rates have been observed in tumors with point mutations and rearrangements [8]. As in adults, more aggressive cancer behavior is expected in cases harboring p.V600E or other BRAF point mutations or BRAF-like fusions (e.g., RET/PTC1, NTRK3/ETV6, ALK rearrangements), although these correlations remain unconfirmed in children [46]. Among the genetic alterations encountered in thyroid cancers of adults, TERT promoter (pTERT) mutations, particularly p.C288T and p.C250T, have shown an incidence of 5–25% in some series. This data is confirmed by The Cancer Genome Atlas cohort with 9.4% of these cancers harboring pTERT mutations, none of them being pediatric. The relatively low frequency of pTERT mutations in children (<5%) has been confirmed by different studies that failed to demonstrate such alterations in different series [47,48,49]. Exceptions are represented by sporadic reports demonstrating isolated cases with p.C288T mutation in PTC or even rarer studies showing slightly higher prevalence (27%), always in restricted cohorts [43,50]. As per pTERT mutations, the presence of highly aggressive mutations in TP53 are quite rare in pediatric population [8]. An indeterminate FNA cytological result combined with a less aggressive molecular variant, such as RAS, DICER1, PTEN, TSHR mutations or PAX8-PPARG rearrangements, may also prompt a conservative surgical approach in children too [2]. The pathogenic role of some of these low-risk mutations is not fully understood, and the association of copy number variation (CNV) is suspected to be necessary for malignant transformation [51]. According to Baran et al., low- and high-risk mutations account for approximately 50% and 30% of malignant pediatric nodules, respectively, while the remaining cases harbor relatively uncommon variants (APC, BLM, PPM1D mutations, FGFR1 rearrangements) which require further investigation [46]. In addition, they observed a correlation between the Bethesda category and the presence of high-risk genetic alterations, opening the possibility of creating an effective cytological and molecular combined algorithm of risk classification, as previously performed with ultrasound and FNA [1,19,46]. DICER1 mutations seem to correlate with the macrofollicular variant of FTC, in which have been found in up to 50% of cases, and seem to be involved in the progression from FTC to PDTC [10]. PTC with solid pattern of growth seems to be correlated with NTRK fusions instead [10]. Interestingly, in pediatric population has been found a higher prevalence of STRN-ALK rearrangements (7% vs. 3%, respectively), both in FTC and PTC, in classic papillary and in diffuse sclerosing variants [10]. Molecular analysis is also pivotal for the diagnosis and management of medullary thyroid carcinoma (MTC), a rare type of thyroid cancer (3–5% of cases) that originates from parafollicular C cells. MTC is associated with activating mutations in the RET proto-oncogene, usually in exon 10 or 11, and different RET mutations correlate variably with disease progression and response to therapy. Patients with germline mutations in RET have a higher risk of developing MTC and may require prophylactic thyroidectomy; therefore, pediatric MTC prompts the evaluation of syndromic association. Molecular assessment is useful also for the application of targeted therapies that specifically inhibit the RET signaling pathway in advanced or metastatic MTC, such as vandetanib and cabozantinib [10].

In summary, molecular analysis of thyroid nodules, especially thyroid FNA specimens, is a valuable tool in the management of thyroid lesions, providing important diagnostic and prognostic information that can guide clinical decision-making and improve patient outcomes.

### 3.4. Future Perspectives

Artificial intelligence (AI) may play a significant role in the analysis of ultrasound and other imaging data (radiomics). Indeed, complex algorithms can automate the process of interpreting images, allowing for faster and more accurate diagnoses even in this field, as well as for the prediction of molecular alterations (Figure 2) [52].

According to a recent meta-analysis, computer-assisted diagnosis has proven to have reliable diagnostic performance, comparable to that of radiologists [53]. A machine learning approach to identify pediatric thyroid nodules requiring surgical intervention demonstrated good performance, but additional data on larger cohorts are needed before their application in clinical practice [54]. These tools can also be trained to autonomously identify and classify cells based on their morphological features and so to detect cancer cells. A study used a combination of clinical data, microscopic characteristics obtained from FNA and imaging features to triage indeterminate and follicular lesions into high- or low-risk categories, reaching a diagnostic accuracy comparable to that of TI-RADS [55]. Currently, in clinical practice, there are no reliable methods for analyzing cells directly from FNA specimens owing to the complex and multi-layered nature of these samples, as well as the lack of standardized algorithms [52]. Even if the benefits and possible challenges of an AI approach are currently well known, the same is still not true for their pediatric counterparts, in which AI and machine learning still represents an emerging field, and the development of useful candidate algorithms to be applied on histology/cytology for computer-aided diagnosis are mainly limited by the relative rarity of these tumors and their intrinsic structural heterogeneity [56]. Another approach may be the application of Matrix-Assisted Laser Desorption/Ionization—Mass Spectrometry imaging (MALDI-MSI) technology to investigate the spatial distribution of biomolecules (proteins, lipids and metabolites) directly in situ, potentially capable of performing a triage of pediatric thyroid nodules [57]. This technology has been previously used to investigate proteomic data of adult indeterminate nodules, NIFTPs and Hashimoto thyroiditis starting from FNA specimens [58,59,60,61].

## 4. Conclusions

It is important to prioritize the development of treatment guidelines and risk stratification tools to accurately identify pediatric patients who would benefit from surgical intervention. In the future, the potential combination of all clinical, radiological, cytological and molecular data merged with the assistance of artificial intelligence will certainly improve thyroid nodule management and surveillance, and will aid in the understanding of the key biomolecular pathways involved in thyroid oncogenic processes. In this direction, multicentric efforts and collaborations can further help in increasing the number of case series describing uncommon clinical entities and molecular alterations in this group of patients, enhancing our capabilities of developing robust and reliable AI tools for their assessment.

## Figures and Tables

**Figure 1 cancers-15-02039-f001:**
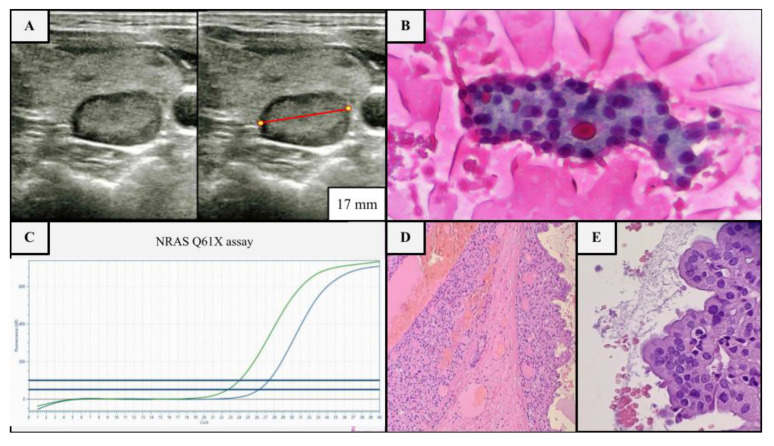
Example of US, cytological and molecular assessment of a thyroid nodule in a 17-year-old female patient with multinodular goiter without clinical risk factors. (**A**) Nodular thyroid lesion of the right lobe, 17 mm in diameter (red line), isoechoic, with a mixed composition and smooth margins, no microcalcifications, deserving FNA according to ATA guidelines (≥1 cm partly solid nodule), but not deserving it according to ACR TI-RADS (class 2) (**B**) FNA shows a blood-rich background with fluid colloid, rare foamy histiocytes and numerous groups of thyroid cells mainly arranged in sheets and microfollicular structures, sometimes with oxyphilic appearance (class IV sec. TBSRTC, Papanicolau stained smear, ×40). (**C**) Molecular test (real time PCR) was performed, since the indeterminate cytology, starting from the needle wash, revealing the NRAS p.Q61X mutation (blue line mutated allele, green line wild type allele), a low-risk mutation group, which confirms the clonality of the lesion and its likely low biological aggressiveness. (**D**,**E**) Accordingly, a hemi-thyroidectomy was performed, revealing a partly cystic nodule composed of oncocytic thyrocytes, with pushing borders and a thin fibrotic capsule without invasion or lymphovascular infiltration, leading to the final histological diagnosis of NRAS-mutant oncocytic follicular adenoma (H & E, 10× and 40×).

**Figure 2 cancers-15-02039-f002:**
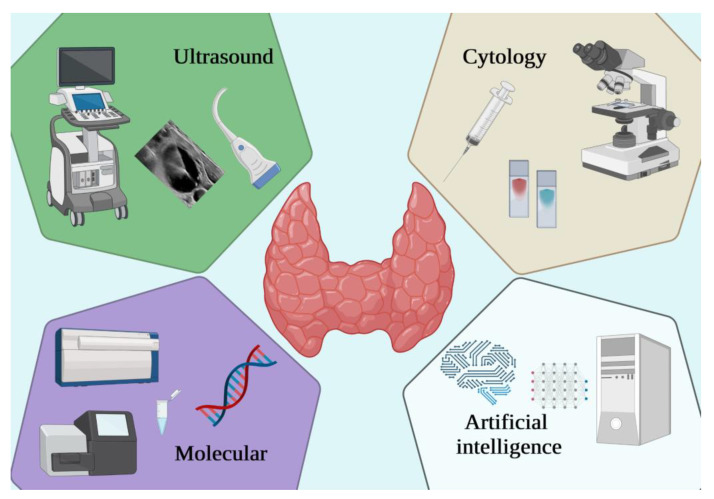
Putative future combined approach for correct identification of malignant thyroid nodules.

**Table 1 cancers-15-02039-t001:** Search strategy summary.

Items	Specifications
Date of Search	1 January 2023
Databases and other sources searched	PubMed (Medline)
Search terms used	Search terms: “thyroid FNA” and “pediatric(s)” or “children”
Timeframe	From 1 January 2020 to 31 December 2022
Inclusion and exclusion criteria	Inclusion criteria:
	(1) study was written in English language;
	(2) the full article could be obtained.
	Exclusion criteria:
	(1) study was written in non-English language;
	(2) the full article was not available;
	(3) study was not related to thyroid FNA in pediatrics;
	(4) study was not published in a peer-reviewed journal.
Selection process	The literature review and the data extraction were conducted independently by two reviewers (D.S. and F.P.).
	A secondary search of the literature was manually conducted from the references of our primary search included papers by the application of the same inclusion and exclusion criteria.
	Doubts or disagreements regarding the inclusion or exclusion of manuscripts were resolved through a discussion between the reviewers until a consensus was reached.

**Table 2 cancers-15-02039-t002:** US suspicious features and clinical risk factors prompting FNA assessment of pediatric thyroid nodules less than 1 cm in size according to the ATA guidelines [2,15].

US Suspicious Features	Clinical Risk Factors
Irregular shape (taller than wide) Irregular margins Micro/macrocalcifications Hypoechogenicity Solid composition Hypervascularity Extrathyroidal extension Abnormal lymph nodes	Autoimmune thyroiditis (Hashimoto thyroiditis, Graves’ disease) Prior radiation (peak incidence between 15 and 30 years later) Prior chemotherapy with alkylants Genetic cancer predisposition syndromes: Familial Adenomatosis Polyposis (FAP, APC-mutated), Carney complex (CC, PRKAR1A-mutated), DICER1 syndrome, PTEN Hamartoma Tumor Syndrome (PHTS, including Cowden syndrome), Multiple Endocrine Neoplasia syndromes (MEN), Werner syndrome (WRN-mutated), Pendred syndrome (SLC26A4-mutated), and others Family history of thyroid cancer Iodine deficiency

**Table 3 cancers-15-02039-t003:** ACR TI-RADS, EU-TIRADS and ATA guidelines FNA flowcharts for the adult population compared to ATA recommendations for pediatrics [15,16,17].

Adults						Pediatrics
ACR Classes	Final Indication ACR	EU Classes	Final Indication EU	ATA Classes	Final Indication ATA	ATA
ACR1	No FNA	EU1	No FNA	ATA1	No FNA	No FNA if <1 cm and no US or clinical risk factors
ACR2	No FNA	EU2	No FNA	ATA2	FNA or follow if ≥2 cm	
ACR3	FNA if ≥2.5 cm	EU3	FNA if >2 cm	ATA3	FNA if ≥1.5 cm	FNA if <1 cm and US or clinical risk factors
	Follow if ≥1.5 cm					
ACR4	FNA if ≥1.5 cm	EU4	FNA if >1.5 cm	ATA4	FNA if ≥1 cm	FNA to any nodule ≥1 cm solid/partly solid
	Follow if ≥1 cm					
ACR5	FNA if ≥1 cm	EU5	FNA if >1 cm	ATA5	FNA if ≥1 cm	
	Follow if ≥0.5 cm		Follow if ≤1 cm			

**Table 4 cancers-15-02039-t004:** Thyroid FNA risk of malignancy (ROM) in adults according to current TBSRTC (The Bethesda System for Reporting Thyroid Cytopathology), SIAPEC (Società Italiana di Anatomia Patologica e di Citopatologia diagnostica) and BTA (British Thyroid Association) reporting thyroid cytology classifications [23,25,26].

Adults					
TBSRTC	ROM (%)	SIAPEC	ROM (%)	BTA	ROM (%)
I	1–4	TIR1-TIR1c	nd	THY1-THY1c	4
II	<3	TIR2	<3	THY2-THY2c	1.4
III	5–15	TIR3A	<10	THY3a	17
IV	15–30	TIR3B	15–30	THY3b	up to 40
V	60–75	TIR4	60–80	THY4	up to 68
VI	97–99	TIR5	>95	THY5	up to 100

**Table 5 cancers-15-02039-t005:** Thyroid FNA ROM according to the main pediatric studies using TBSRTC categories [23].

Pediatrics						
Studies	TBSRTC Categories (%)					
	I	II	III	IV	V	VI
Amirazodi et al. [27]	0	16	67	-	71	100
Buryk et al. [28]	0	10	0	50	86	100
Cherella et al. [29]	11	0.7	44	71	73	97
Heider et al. [30]	-	-	36	20	100	-
Jia et al. [31]	0	0.8	15.6	54.5	100	100
Kardelen et al. [32]	0	25	100	75	85.7	100
Lale et al. [33]	17	0	50	47	100	100
Monaco et al. [34]	0	7	28	58	100	100
Norlen et al. [35]	0	0	22	100	100	100
Pantola et al. [36]	0	0	8.3	10	100	100
Partyka et al. [37]	0	1.5	21	57.1	100	100
Rossi et al. [38]	0	0	11.8	81.8	100	100
Wang et al. [39]	0	7	20	25	100	100
Suh et al. [40]	22	3.2	75	50	100	100
Tuli et al. [21]	0	0	0	77.8	100	100
Vuong et al. [41]	30	50	66.6	36.4	100	99.3
Average	5.3	10.4	38.5	50.6	95.3	99.8

**Table 6 cancers-15-02039-t006:** Molecular signatures of thyroid cancer in pediatric and adult; SNV: single nucleotide variation; pTERT: TERT promoter [10].

Mutation Type	Molecular Landscape	Pediatric	Adult
**PTC**			
SNV	BRAF V600E	<70%	30–90%
	pTERT	<5%	5–25%
	DICER1	5–10%	<5%
	HRAS, KRAS, NRAS	<10%	<35%
Fusion	RET	20–80%	5–35%
	PAX8-PPARG	<5%	<5%
	NTRK	<20%	<5%
	BRAF	<20%	<3%
	ALK	<10%	<3%
**FTC**			
SNV	HRAS, KRAS, NRAS	<50%	10–60%
Fusion	DICER1	<25%	1%
	PTEN	<25%	<1%
	PAX8-PPARG	<20%	35–50%

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
