# Peer review of "The Complex Cyto-Molecular Landscape of Thyroid Nodules in Pediatrics"

_cancers, 2023, doi:10.3390/cancers15072039_

Round 1
Reviewer 1 Report
The authors reviewed the cyto-molecular features of thyroid nodules in pediatrics from the viewpoint of clinical diagnosis by ultrasonography. The manuscript is well organized and the contents sounds interesting. I, however, have some concerns as follows.
1. The authors described the molecular testing in pediatrics (line 199-227). They should make a table showing the molecular differences between pediatrics and adults.
2. In p.7, line 216, the authors described mutations in TERT genes. They should specify the mutations. TERT promoter mutations are related to the poor prognosis in thyroid cancer. In addition, single nucleotide polypmorphisms (SNPs) of TERT are recently implicated in the pathogenesis of thyroid cancer. The authors should review TERT mutations in detail.
3. “Future perspectives” (p.7, line 229) might be “3.4. Future perspectives”.
4. The authors described the computer-assisted diagnosis (p.7, line 233-). Are there any difference in the computer-assisted diagnosis between pediatrics and adults? Are there any reports describing the problems of AI-assisted diagnosis in pediatrics?
Reviewer 2 Report
This is an interesting narrative review, which is somewhat hampered by its use of English. Furthermore, it lacks a "punchline" /take home message. Is there any novelty and/or personal contribution of the authors?
Reviewer 3 Report
Authors have meticulously compiled the available information on the recent advances in the diagnosis and management of thyroid nodules in children. The review is extensive; and will serve as an excellent source of information on the topic. It provides extended information on the use of thyroid fine needle aspiration, ultrasound assessment, molecular testing, and the great potential of AI. I only have a few minor observations:
1. There are some typos and grammatical errors throughout the manuscript. The work would benefit from close editing.
2. Avoid using figures in the conclusion section. Please relocate it.
